# The core outer junction protein CFAP77 connects A- and B-tubules within doublet microtubules of cilia and flagella

Lan Xia[1☉], Guo-Liang Yin[2,3☉], Yu Long[1], Fei Sun[2,3,4,5], Bin-Bin Wang[6,7,8*], Yun Zhu[2*], Su-Ren Chen[1*]

**1** Key Laboratory of Cell Proliferation and Regulation Biology, Ministry of Education, Department of Biology, College of Life Sciences, Beijing Normal University, Beijing, China, **2** Key Laboratory of Biomacromolecules, CAS Center for Excellence in Biomacromolecules, Institute of Biophysics, Chinese Academy of Sciences, Beijing, China, **3** School of Life Sciences, University of Chinese Academy of Sciences, Beijing, China, **4** Center for Biological Imaging, Institute of Biophysics, Chinese Academy of Sciences, Beijing, China, **5** Guangzhou Institutes of Biomedicine and Health, Chinese Academy of Sciences, Guangzhou, Guangdong, China, **6** Center for Genetics, National Research Institute of Family Planning, Beijing, China, **7** Graduate School of Peking Union Medical College & Chinese Academy of Medical Sciences, Beijing, China, **8** NHC Key Laboratory of Reproductive Health Engineering Technology Research (NRIFP), National Research Institute for Family Planning, Beijing, China

☉ These authors contributed equally to this work.
* wbbahu@163.com (BW); zhuyun@ibp.ac.cn (YZ); chensr@bnu.edu.cn (SC)

## Abstract

The assembly and physiological function of cilia and flagella depend on the stable association of A- and B-tubules, which form axonemal doublet microtubules (DMTs). However, the mechanisms underlying the connection of B-tubules to A-tubules to form DMTs in mammalian cilia/flagella are unclear. *CFAP77* encodes an outer junction (OJ) protein within DMTs that is conserved across many species and cell types. In this study, *Cfap77*-KO mice were generated to reveal that CFAP77 is essential for sperm progressive motility and male fertility. Loss of CFAP77 led to opened B-tubules specifically at the OJ regions of axonemal DMTs as revealed by conventional transmission electron microscopy. Cryo-electron tomography was used to further resolve the in situ structure of sperm axonemal DMTs directly from *Cfap77*-KO mice, which exhibited a loss of large filamentous density corresponding to the CFAP77-CCDC105-TEX43 ternary subcomplex at the OJ regions. Additionally, sperm proteomic analysis confirmed that CFAP77 knockout led to the complete loss of this ternary complex. Our work not only explores the physiological role of the OJ protein CFAP77 in axonemal A- and B-tubule connections in mammals but also combines in situ structural biology and knockout mice to reveal the underlying structural/molecular mechanism involved.

**Data availability statement:** All relevant data are within the paper and its Supporting information. Proteomics data have been deposited to the ProteomeXchange Consortium using the iProX partner repository with the dataset identifier PXD056128. The map of sperm DMTs from *Cfap77*-KO mice has been deposited in the EMDB under accession code EMD-63176568.

**Funding:** This work was supported by the National Natural Science Foundation of China (32370905 and 32571005 to S.C.), Basic Research Projects of Central Scientific Research Institutes (2023GJZD01 to B.W.), Fundamental Research Funds for the Central Institutes (2023GJZD01 to B.W.), National Natural Science Foundation of China (32471244 to Y.Z.), Beijing Natural Science Foundation (JQ24056 to Y.Z.), National Key Research and Development Program (2021YFA1301500 to F.S.), National Key Research and Development Program (2024YFA1307402 to Y.Z.), and Open Fund of Key Laboratory of Cell Proliferation and Regulation Biology, Ministry of Education (to S.C.). The funders had no role in study design, data collection and analysis, decision to publish, or preparation of the manuscript.

**Competing interests:** The authors have declared that no competing interests exist.

**Abbreviations:** CPC, central pair complex; DMT, doublet microtubules; IJ, inner junction; OJ, outer junction; MIP, microtubule inner protein; cryo-EM, cryo-electron microscopy; cryo-ET, cryo-electron tomography; PDB, protein data bank; co-IP, co-immunoprecipitation; WT, wild-type; KO, knockout; IVF, in vitro fertilization; TEM, transmission electron microscopy; cryo-FIB, cryo-focused ion beam thinning; STA, sub-tomogram averaging; SEM, mean±standard error; MIVA, MIP-variant-associated asthenozoospermia; GA, glutaraldehyde; PB, phosphate buffer.

## Introduction

Cilia protrude from the surface of almost all eukaryotic cells and play many diverse roles in sensory processes, motility, and human physiology. The core structure of motile cilia, the axoneme, consists of a pair of microtubule singlets termed the central pair complex (CPC) and nine peripherally positioned doublet microtubules (DMTs), which form a '9+2′ architecture [1,2]. Sperm are a type of cell with a specialized function and morphology, and their motility is critical for fertilization [3]. The '9+2′ structure is shared by motile cilia and sperm flagella, with some differences between species and tissues.

Each DMT has a distinctive structure with a complete ring of 13 protofilaments (the A-tubule) and an incomplete ring of 10 protofilaments (the B-tubule) [1]. DMTs are docked with a set of axonemal complexes such as outer and inner dynein arms, radial spokes and a nexin-dynein regulatory complex [2]. The connection of B-tubules to A-tubules is believed to be critical for axonemal stability and cilia/flagellar motility. A long-standing question is how B-tubules attach tightly to A-tubules within DMTs.

Inner junction (IJ) proteins at A01-B10 and outer junction (OJ) proteins at A11-A12 and B01-B02 are suggested to mediate this connection [1,2,4]. Recently, we and others determined the high-resolution structure of human/mouse/bovine sperm axonemal DMTs, revealing many core and sperm-specific microtubule inner proteins [5–10]. CFAP20, PACRG, ENKUR, CFAP52 and other proteins are core components of the IJ, whereas CFAP77 and other proteins are localized at the OJ regions [5–12]. The expression/distribution of these genes is conserved in cilia/flagella across many species, including *Chlamydomonas*, *Tetrahymena*, mice, bovines, and humans [4–13]. Despite their important positions, their physiological roles in IJ/OJ structure and cilia motility are largely unknown. CFAP20 has been reported to maintain the structural integrity of the IJ and is required for cilia motility in zebrafish as well as nonmotile cilia function in *C. elegans* [14]. Knockout of CFAP77 moderately reduces *Tetrahymena* cell motility and partially destabilizes the OJ structure [13]. However, their effects on mammalian cilia/flagella function and the underlying structural/molecular mechanisms of IJ/OJ stability are still unclear.

In this study, we generated a *Cfap77*-KO mouse strain and found that male knockout mice were infertile, with defects in sperm motility. Intriguingly, loss of CFAP77 led to a disconnection of A- and B-tubules, specifically at the OJ regions. Cryo-ET was used to further resolve the in situ structure of sperm axonemal DMTs in *Cfap77*-KO mice, which exhibited a loss of large filamentous density corresponding to the CFAP77-CCDC105-TEX43 ternary subcomplex at the OJ regions. The absence of this ternary complex represents an initial event underlying the disconnection of A- and B-tubules at OJ sites, axoneme instability, impaired sperm progressive motility, and male infertility in *Cfap77*-KO mice.

## Results

### CFAP77-CCDC105-TEX43 subcomplex at the OJ regions of sperm DMTs

Recently, we and others resolved the near-atomic-level structure of DMTs from mouse sperm [6–9], which provides a valuable source for further explorations of the

functions of the individual DMT structural components. CFAP77, CCDC105, and TEX43 attracted our attention because they located at OJ regions (PDB entry 8IYJ), which may play important roles in the connection of B-tubules to A-tubules within sperm DMTs (Fig 1A and 1B). In mouse sperm, CFAP77 binds to A11-A12 and B01-B02 protofilaments in a 16 nm repeat. CCDC105 resides between protofilaments A11 and A12, assembling into a long filament by interacting with neighbouring CCDC105 molecules. CCDC105 filaments seemingly contact individual CFAP77 proteins into a stable structure to connect A- and B-tubules. TEX43 is located at the junction site of two CCDC105 proteins, suggesting that it plays a role in the reinforcement of CCDC105 filaments (Fig 1A-1C).

There are many and close interactions among CFAP77, CCDC105 and TEX43 [7,9] (Fig 1C). Specifically, CCDC105 features two helical bundles: one bundle binds to the glutamine (Q)208-alanine (A)269 region of CFAP77, while the other binds to the leucine (L)132-L140 region of CFAP77 (Fig 1C). Additionally, the proline (P)59-lysine (K)107 region of TEX43 interacts with the interface of two adjacent CCDC105 proteins (Fig 1C). To validate the interactions in this ternary subcomplex, we performed coimmunoprecipitation (co-IP) analysis and found that CCDC105 interacted with both CFAP77 and TEX43 but that there was no interaction between CFAP77 and TEX43 (S1 Fig). We further mutated the key residues at one of the interaction interfaces between CFAP77 and CCDC105. The methionine (M)352-glycine (G)355 residues of CCDC105 were mutated to alanine (A)AAA, and the G131-valine (V)133 residues of CFAP77 were mutated to AAA. Co-IP analysis revealed that the interaction between CFAP77 and CCDC105 was obviously attenuated after mutation of those critical contact sites (Fig 1D and 1E). Together, these results suggest that the CFAP77-CCDC105-TEX43 subcomplex at the OJ regions may be critical for DMT A- and B-tubule connection and sperm motility.

### *Cfap77*-KO male mice are infertile and produce sperm with motility defects

Knockout of CFAP77 moderately reduces *Tetrahymena* cell motility and partially destabilizes the OJ [13]. To reveal the physiological role of CFAP77 in mammals, we generated a *Cfap77*-KO mouse strain via CRISPR/Cas9 technology. Exons 2 and 3 of the *Cfap77*–202 transcript (ENSMUST00000157048) were selected as the knockout region to generate a 30,988 bp deletion (Fig 2A). Western blot analysis revealed CFAP77 expression to be completely absent in the sperm samples of knockout mice (Fig 2B). *Cfap77*-KO mice were healthy and developed normally; however, fertility tests revealed that *Cfap77*-KO male mice were infertile (Fig 2C). In vitro fertilization (IVF) assays using cumulus-intact oocytes (Fig 2D) or cumulus-free oocytes (Fig 2E) revealed that the percentage of 2-cell embryos was significantly lower in the groups with sperm from *Cfap77*-KO mice than in those with sperm from wild-type (WT) mice. However, the percentage of 2-cell embryos generated from *Cfap77*-KO sperm with zona pellucida-free oocytes was comparable to that of the control (Fig 2F), indicating that CFAP77 loss does not affect sperm–egg fusion or genomic integrity. Histological examination of the testes and cauda epididymis of *Cfap77*-KO mice further revealed a normal process of spermatogenesis and sperm production (S2 Fig). We next examined semen characteristics. Papanicolaou staining revealed that the morphology of sperm from *Cfap77*-KO mice was generally normal (Fig 2G). The sperm count was also similar between *Cfap77*-KO mice and WT mice (Fig 2H), whereas progressive sperm motility was significantly attenuated in *Cfap77*-KO mice (Fig 2I). Through sperm motion tracking and statistical analysis, we found that the percentage of sperm showing S-shaped, O-shaped, and shake/silent movements was significantly greater in *Cfap77*-KO mice than in WT mice (Fig 2J and S1 Video). Collectively, these findings indicate that CFAP77 is critical for progressive sperm motility and male fertility in mice.

### Open DMT B-tubules at the OJ regions in the sperm of *Cfap77*-KO mice

To investigate the structural basis for impaired progressive sperm motility in *Cfap77*-KO mice, we examined the ultrastructure of the sperm flagellar axoneme by traditional transmission electron microscopy (TEM). The ultrastructures of the acrosome, CPC, radial spokes, dynein arms, and accessory structures of axonemes (e.g., the mitochondrial sheath and fibrous sheath) were indistinguishable between *Cfap77*-KO mice and WT mice (S3A Fig). The disconnection of A- and B-tubules within DMTs represented as the only obvious ultrastructural defect after the loss of CFAP77 (Fig 3). The

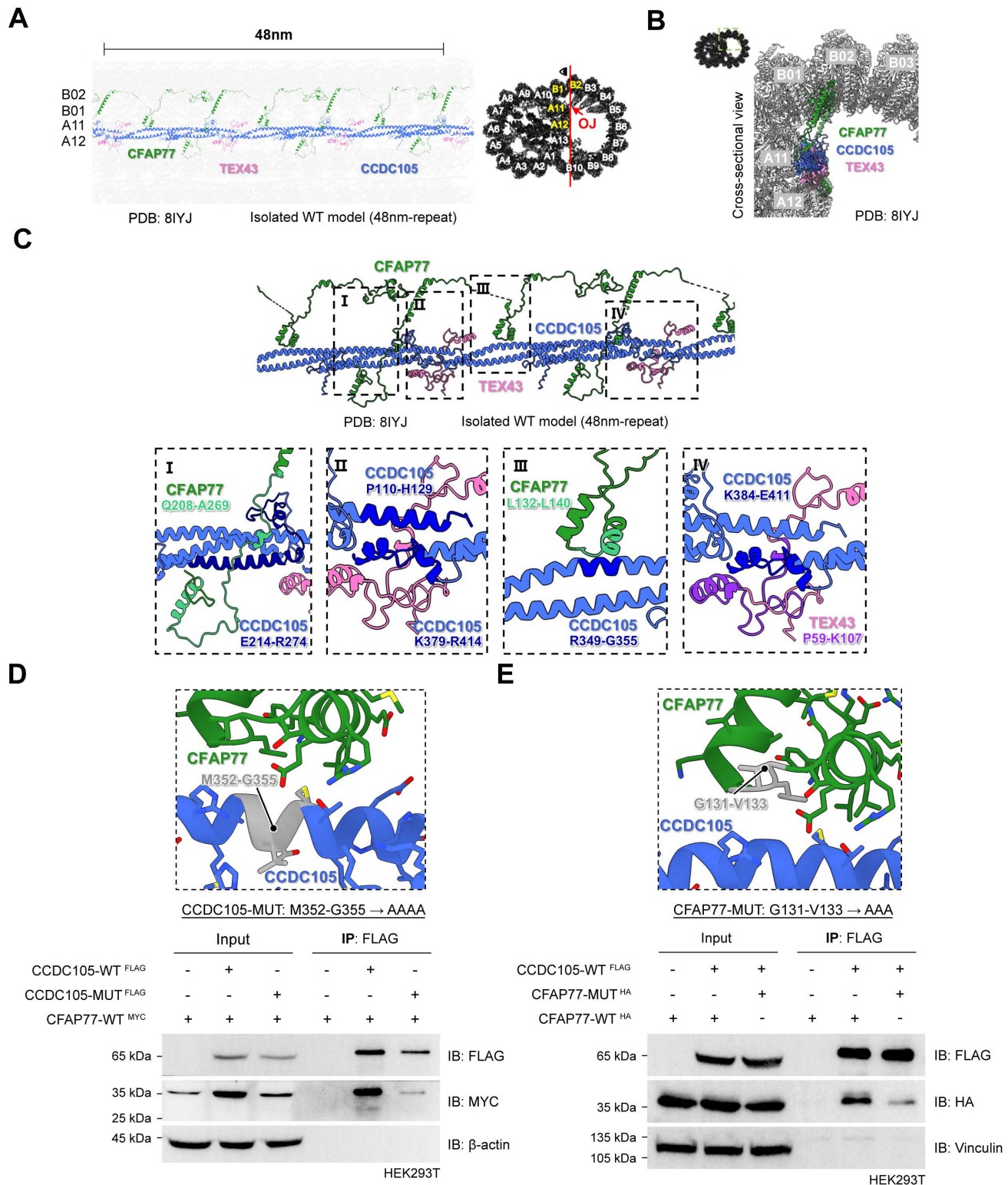

**Fig 1. CFAP77-CCDC105-TEX43 subcomplex at the OJ sites of the sperm axoneme. (A, B)** Structural models of the CFAP77-CCDC105-TEX43 subcomplex at the OJ regions of the mouse sperm axoneme are shown with a 48 nm length of vertical section in the isolated DMT (PDB: 8IYJ). CFAP77 resided between A11-A12 and B01-B02. CCDC105 and TEX43 were localized at A11-A12. OJ, outer junction. **(C)** The interaction interfaces of

the CFAP77-CCDC105-TEX43 subcomplex are shown for the isolated mouse sperm DMT model (PDB: 8IYJ). The interaction interfaces of CFAP77-CCDC105 **(I)**, CCDC105-CCDC105 **(II)**, CCDC105-CFAP77 (III) and CCDC105-TEX43 (IV) are enlarged. Residues at the interaction interfaces were labelled. **(D)** Coimmunoprecipitation of MYC-tagged CFAP77 by FLAG-tagged WT-CCDC105 and mutant (MUT)-CCDC105 in HEK293T cells. For the CCDC105 mutant plasmid, amino acids 352-355 were mutated into AAAA. β-actin served as the internal control. **(E)** The ability of FLAG-tagged CCDC105 to interact with HA-tagged WT-CFAP77 or MUT-CFAP77 in HEK293T cells. For the CFAP77 mutant plasmid, amino acids 131-133 were mutated into AAA. Vinculin served as the internal control. Raw blot images can be found in S1 Raw Images.

sperm from the WT mice presented well-organized "9+2" axonemes, with DMT B-tubules closely attached to the DMT A-tubules (Fig 3A). In contrast, a large proportion of sperm in *Cfap77*-KO mice presented defects, specifically at the A- and B-tubule connection sites (open DMT B-tubules at the OJ regions) (Fig 3B). In the axonemes of *Cfap77*-KO mice, the disconnection of A- and B-tubules was distributed mainly to DMTs 1, 4, 5, 6, and 9 (S3B Fig). Statistical analysis revealed that the percentage of axonemes with open DMT B-tubules significantly increased after the loss of the CFAP77 protein (6.67% ± 1.453% in WT mice and 45.33% ± 3.480% in *Cfap77*-KO mice) (Fig 3C). In *Cfap77*-KO mice, this ratio was not obviously different between the mid-piece and the principal piece of the sperm flagella (S3C Fig). Intriguingly, the percentage of disconnection of A- and B-tubules was further increased in *Cfap77*-KO mice after sperm capacitation for 1 hour (72.00% ± 4.141%), suggesting that the instability of axonemes increased during sperm hyperactivation (Fig 3C). The phenotype of open DMT B-tubules specifically at the OJ regions in *Cfap77*-KO mice is consistent with the structural data from WT DMTs showing that CFAP77 is selectively localized at the OJs of DMTs. Collectively, the ultrastructural analysis clearly revealed that detachment of A- and B-tubules within DMTs is the underlying reason for the disrupted progressive motility of sperm in *Cfap77*-KO mice.

### Insight into the in-cell structure of sperm DMTs in *Cfap77*-KO mice

To investigate the impact of CFAP77 deletion on the structural assembly of sperm DMTs at a relatively high resolution, we froze sperm from *Cfap77*-KO mice on a grid and milled the sample to a thickness of approximately 200 nm via cryo-focused ion beam thinning (cryo-FIB) (S4A Fig). We collected cryo-ET data targeting the sperm tail and resolved the in situ structure of DMTs using a subtomogram averaging (STA) approach (S4B and S5 Figs). During particle picking and data processing, we found that DMT particles presented notable structural heterogeneity, indicating that the ultrastructures of the sperm DMTs in *Cfap77*-KO mice were compromised. This finding is consistent with our conventional TEM observations that approximately 45% of DMT B-tubules were opened at the OJ regions in sperm from *Cfap77*-KO mice (Fig 3). After discarding many particles through 3D classification, we obtained the 8-nm repeat structure of DMTs with a resolution of 24 Å (Fig 4). The 8-nm repeat WT difference map was generated from the reported 16-nm repeat WT map (EMD-35210/35211). It was then resampled onto the same grid (6.8 Å/pixel), and a low-pass filter was applied to 24 Å, matching the *Cfap77*-KO reconstruction. The difference map was then created by subtracting the density the *Cfap77*-KO map from that of the 8-nm repeat WT map. The difference map confirmed that the major missing part of the DMT structure in *Cfap77*-KO mice was a long fibrous density extending longitudinally along the tubulin wall of the ribbon near protofilament A11 (Fig 4A). Then, we fitted the atomic models of the sperm DMT structure of WT mice to the density map of the sperm DMTs of *Cfap77*-KO mice. Notably, CFAP77, CCDC105, and TEX43 were identified as missing densities in the sperm DMT B-tubules of *Cfap77*-KO mice (Fig 4B). The corresponding density of CFAP77 was not affected by the knockout of the *Cfap77* gene, as expected, but the loss of CCDC105 and TEX43 highlights the essential role of CFAP77 in the formation of the CFAP77-CCDC105-TEX43 ternary complex. Moreover, we applied a soft mask that enclosed only the A-tubules and ignored the density from the B-tubules during initial alignment for all the DMT particles. We then applied a soft mask that enclosed only the B-tubules for 3D classification. Approximately 37% of the particles clustered into the canonical intact DMT class, whereas one class (~12%) clearly displayed an OJ-open architecture (Fig 4C). We also applied a small mask around the OJ for the good DMT particles. This process revealed two subclasses that exhibit weakened

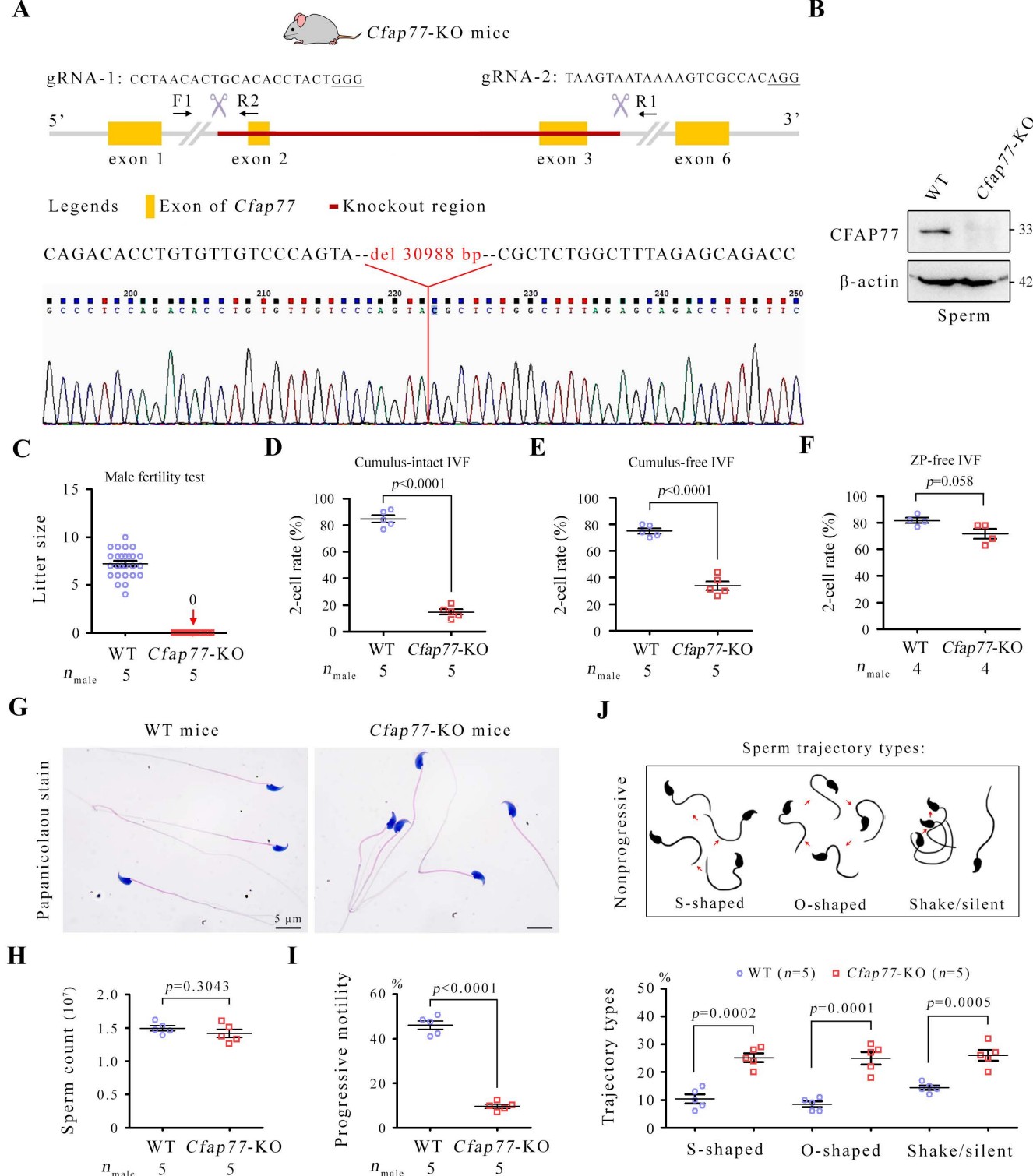

**Fig 2. *Cfap77*-KO mice were sterile due to defective sperm motility. (A)** Genomic features and knockout strategy for mouse *Cfap77* by CRISPR/ Cas9 technology. A 30,988 bp deletion (targeting exons 2-3) in the *Cfap77* gene was confirmed by Sanger sequencing. **(B)** western blot analysis of CFAP77 protein expression in sperm samples from WT and *Cfap77*-KO mice. β-actin served as an internal control. **(C)** The litter size between the WT

male group and the *Cfap77*-KO male group mated with WT female mice was compared. The error bars represent the SEM (*n* = 5). Statistical analysis was not performed because all the data for the *Cfap77*-KO male groups were 0. **(D)** The ratio of 2-cell embryos of cumulus-intact oocytes generated from WT and *Cfap77*-KO mouse-derived sperm (2 × 10$^6$ sperm/mL). Student *t* test; error bars represent the SEM (*n* = 5). **(E)** The ratio of 2-cell embryos generated from WT or *Cfap77*-KO sperm (2 × 10$^6$ sperm/mL) and cumulus-free eggs. Student *t* test; error bars represent the SEM (*n* = 5). **(F)** The ratio of 2-cell embryos of the zona pellucida (ZP)–free in vitro fertilization (IVF) using WT and *Cfap77*-KO mouse-derived sperm (2 × 10$^5$ sperm/mL). Student *t* test; error bars represent the SEM (*n* = 4). **(G)** Papanicolaou staining of sperm from WT mice or *Cfap77*-KO mice. Scale bars, 5 μm. **(H)** Sperm counts of WT mice and *Cfap77*-KO mice. Student *t* test; error bars represent the SEM (*n* = 5). **(I)** Progressive motility of sperm from WT mice and *Cfap77*-KO mice. Three fields of view were randomly selected, and at least 200 sperm from each sample were analysed. Student *t* test; error bars represent the SEM (*n* = 5). **(J)** Schematic diagram illustrating three major types of sperm trajectories showing nonprogressive motility. The percentage of sperm showing S-shaped, O-shaped, and shake/silent types of motility between WT mice and *Cfap77*-KO mice. One hundred sperm were counted for each mouse. Student *t* test; error bars represent the SEM (*n* = 5). The data underlying the graphs shown in the figure can be found in S1 Data. Raw blot images can be found in S1 Raw Images.

protofilament contacts at distinct positions: an A10–B01 weak-connection state (~20%) and a B01–B02 weak-connection state (~18%). Together, these subclasses account for ~38% of the dataset, indicating that many seemingly intact DMTs already harbour incipient breaks at the OJ (Fig 4D). Collectively, our in-cell structural study provides direct visual evidence that the loss of the CFAP77-CCDC105-TEX43 subcomplex at the OJ region represents the initial event precipitating open DMT B-tubules of sperm axonemes in *Cfap77*-KO mice.

## The CCDC105 and TEX43 proteins in testes and sperm were lost after CFAP77 deletion

Given that sperm count and morphology are highly similar between *Cfap77*-KO mice and WT mice, mass spectrometry was further applied to quantitatively identify the sperm proteomes of WT and *Cfap77*-KO mice (*n* = 3 each group). A cut-off point of a 2-fold change and a *p*-value (Student *t* test) less than 0.01 were selected as the screening criteria (Fig 5A). Compared with those in the WT group, the CCDC105 and TEX43 proteins were significantly downregulated in the sperm samples of *Cfap77*-KO mice. This proteome discovery is consistent with our in situ structural analysis of sperm DMTs from *Cfap77*-KO mice, which revealed a missing CFAP77-CCDC105-TEX43 subcomplex at the OJ region (Fig 4). Through western blot analysis, we further confirmed that the expression of CCDC105 (Fig 5B) and TEX43 (Fig 5C) was almost undetectable in both testis and sperm samples from *Cfap77*-KO mice. In contrast, the mRNA levels of *Ccdc105* (Fig 5D) and *Tex43* (Fig 5E) were unaltered after the deletion of CFAP77, indicating that CFAP77 affects the expression/stability of CCDC105 and TEX43 at the protein level rather than at the mRNA level.

## Effect of CFAP77 on other cilia tissues in mice

CFAP77 is a widely conserved OJ protein of axonemal DMTs among eukaryotic ciliated organisms [7–9,13,15,16], with orthologues showing similarity in humans, bovines, mice, sea urchins, *Tetrahymena*, and *Chlamydomonas* (Figs 6A and S6). There are no orthologues of CFAP77 in *Trypanosoma brucei* [17,18]. In contrast, CCDC105 and TEX43 specifically serve as CFAP77-interacting proteins in both sea urchin sperm [6] and mammalian sperm [7–9] (Fig 6A). The existing structural models of the CFAP77-CCDC105-TEX43 subcomplex at mouse sperm DMTs, bovine sperm DMTs, and sea urchin sperm DMTs as well as CFAP77 protein at human respiratory DMTs, pig fallopian tube DMTs, pig brain ventricle DMTs, and *Tetrahymena* DMTs, are presented in the S7 Fig. Sperm have single, long flagella and undergo hyperactivation to fertilize oocytes after long-distance migration. Thus, CCDC105 and TEX43 are expected to be needed to further strengthen the stability of OJs within axonemal DMTs in sperm.

*Cfap77* mRNA was highly expressed in mouse testes and epididymis but was also expressed in other types of tissues (S8 Fig). Given that CFAP77 is also expressed in other types of cilia, we further examined the effects of CFAP77 on ependymal cilia, tracheal cilia and nasal cilia in mice. *Cfap77*-KO mice were healthy and presented no identifiable abnormalities in cilia function, including laterality abnormalities, hydrocephalus, or respiratory problems. As revealed by scanning electron microscopy, the morphology of the ciliary layer in the ependyma, trachea, and nasal cavity was

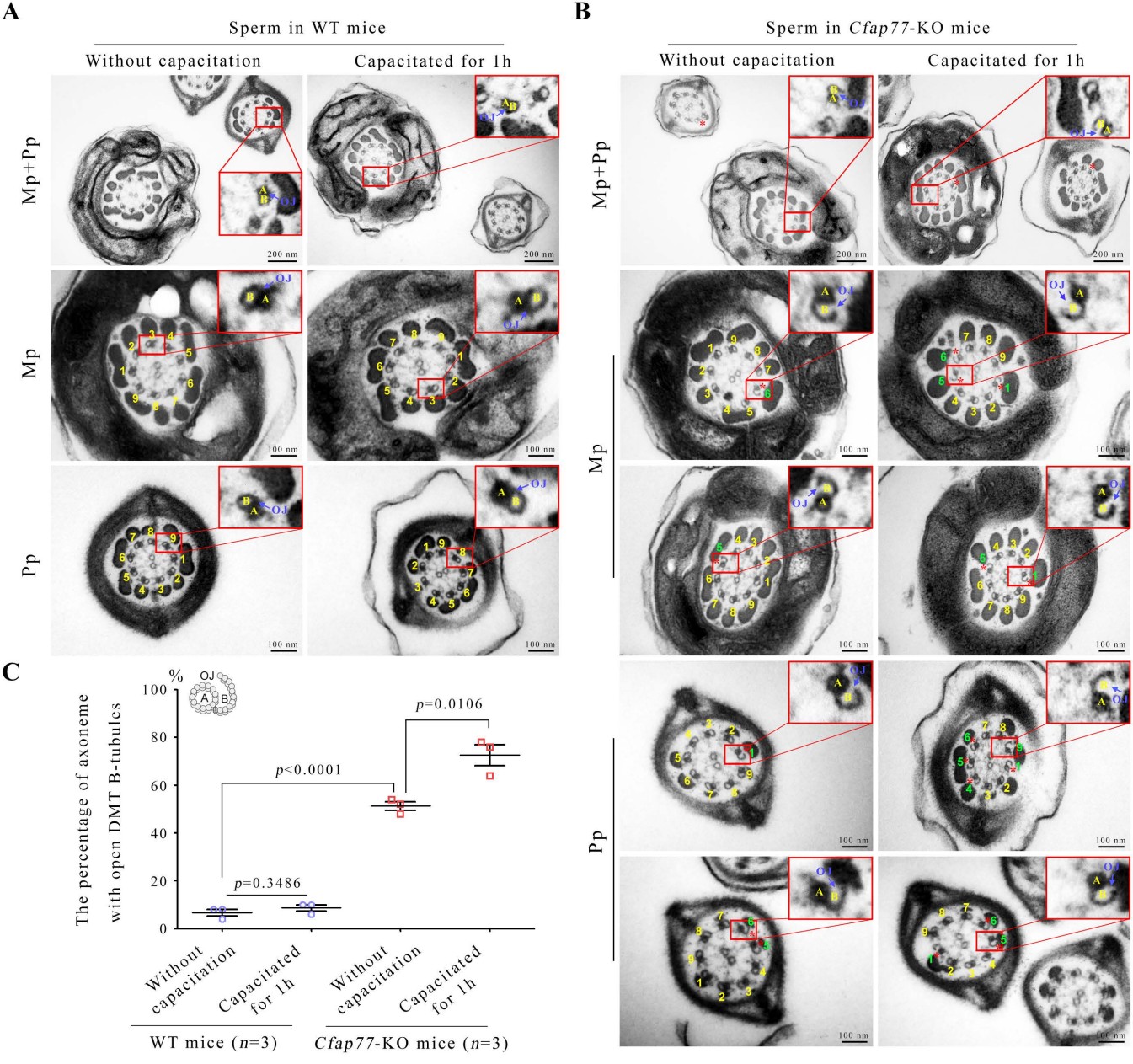

**Fig 3. Open DMT B-tubules of sperm axonemes in *Cfap77*-KO mice. (A)** Transmission electron microscopy (TEM) analysis of the sperm axoneme (9 + 2) from WT mice. Sperm collected from the cauda epididymis were directly fixed or subjected to capacitation in TYH medium for 1 hour and then fixed. Each DMT was numbered. Scale bars, 200 nm or 100 nm. **(B)** TEM images of sperm axonemes from *Cfap77*-KO mice. The principal piece (Pp) and mid-piece (Mp) of the sperm flagella are shown, and a single DMT is magnified. Each DMT was numbered. The green numbers and red stars indicate the OJ regions where open DMT B-tubules exist. Scale bars, 200 nm or 100 nm. **(C)** The percentage of sperm axonemes with open DMT B-tubules was calculated between WT mice and *Cfap77*-KO mice as well as between the uncapacitated and capacitated subgroups. Sperm from the four groups (n = 3 each) were prepared for TEM analysis. For each mouse, 50 axonemes were randomly selected to examine the connection of A- and B-tubules. Student *t* test; error bars represent the SEM (n = 3). The data underlying the graphs shown in the figure can be found in S1 Data.

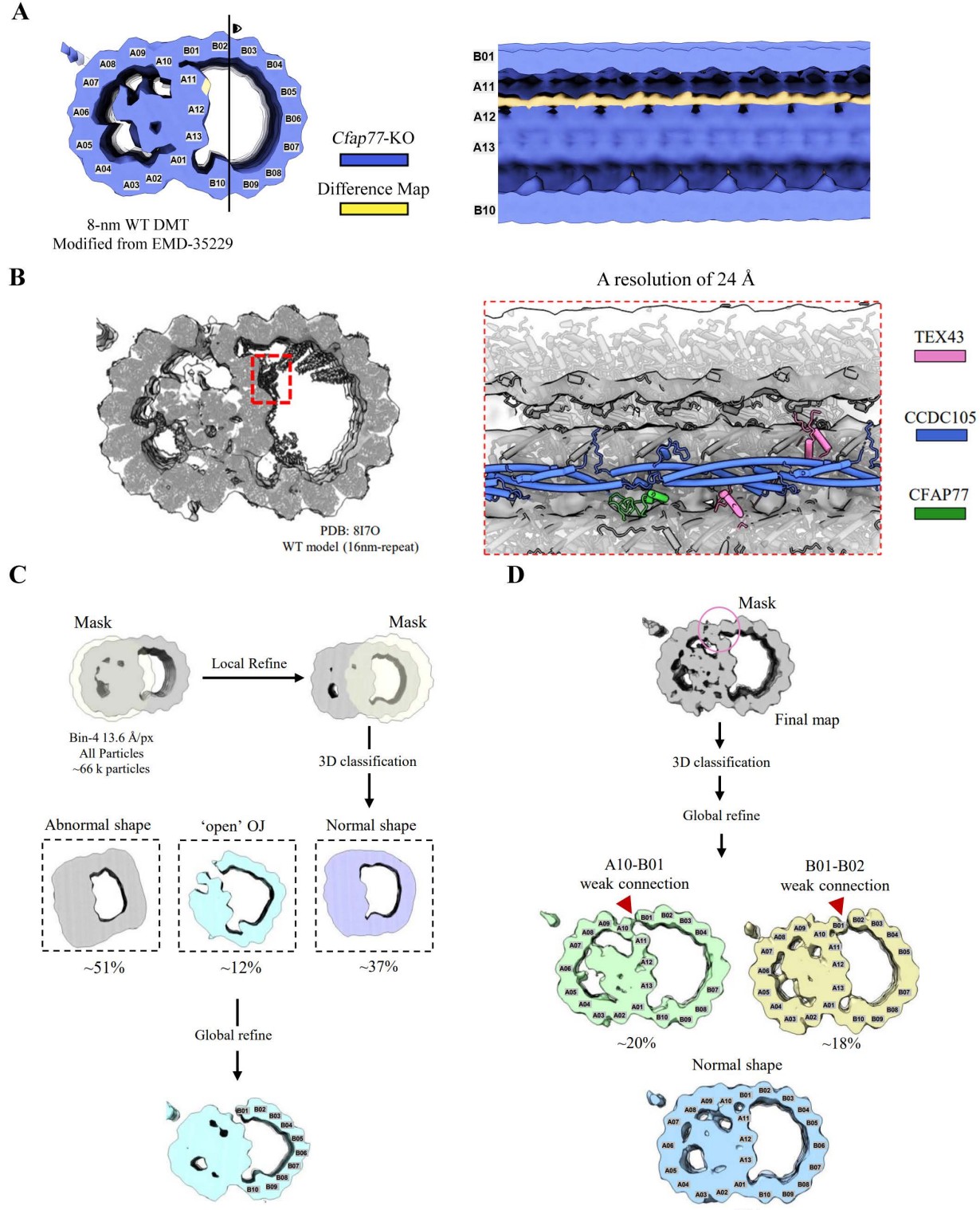

**Fig 4. Overall architecture of sperm axonemal DMTs in *Cfap77*-KO mice. (A)** In-cell structural determination of sperm axonemal DMTs from *Cfap77*-KO mice. A cryo-EM map of DMTs with an 8 nm repeat was obtained by subtomogram analysis. The lost density (yellow colour) in sperm axonemal DMTs from *Cfap77*-KO mice was obtained by subtracting the density map of *Cfap77*-KO DMTs from that of the WT DMTs (EMD-35210/35211).

The transverse sectional view of DMTs and the side view of the B-tubule lumen are shown. **(B)** The atomic model of WT DMTs (PDB: 8I7O) fitted into the density map of *Cfap77*-KO DMTs. The *Cfap77*-KO DMTs presented a loss of density corresponding to the CFAP77-CCDC105-TEX43 subcomplex near protofilaments A11 and A12. **(C)** Focused 3D classification of DMTs in different regions. A soft mask was applied to enclose only the A-tubules, and the density from the B-tubules was ignored during initial alignment for all the DMT particles. A soft mask was then applied to enclose only the B-tubules for 3D classification. The cyan map highlights the subclass that exhibits a pronounced OJ-open fracture. **(D)** OJ-focused classification performed with good DMT particles. Red arrowheads mark the sites at which protofilament contacts are weakened or lost (A10–B01 and B01–B02).

indistinguishable between WT mice and *Cfap77*-KO mice (S9 Fig). Furthermore, the ultrastructures of axonemes in tracheal cilia were examined by the conventional TEM (Fig 6B). The percentage of axonemes with open DMT B-tubules at the OJ regions in tracheal cilia significantly increased (5.33% ± 1.76% in WT mice and 29.33% ± 1.76% in *Cfap77*-KO mice) after the loss of the CFAP77 protein (Fig 6C). However, this percentage of disconnected A- and B-tubules of axonemes in tracheal cilia (less than 30%) was lower than that observed in sperm (approximately 45%) (Fig 3). Taken together, our findings reveal that knockout of the core OJ protein CFAP77 leads to (i) loss of the CFAP77-CCDC105-TEX43 subcomplex in the OJ region, (ii) opening of DMT B-tubules specifically at OJ sites, and (iii) axoneme instability as well as abnormal sperm motility in mice (Fig 6D).

## Discussion

The axoneme, a very large molecular machine, is a cylindrical arrangement of nine DMTs surrounding a CPC. Studying of axoneme organization is critical not only for understanding cilia biology but also for diagnosing ciliopathies. The A- and B-tubules of each DMT are joined at regions termed the OJ and the IJ [4]. One of the key questions in the axoneme field is how the OJ and IJ work to connect A- and B-tubules within DMTs. High-resolution structural analysis of axonemal DMTs and knockout models provides useful information. CFAP77 was identified as a component of the OJ by cryo-EM or cryo-ET [7–9]. Loss of CFAP77 leads to the formation of open DMT B-tubules in *Tetrahymena* [13] and mouse cilia/flagella (this study), suggesting that CFAP77 is an evolutionarily conserved OJ component specifically required for DMT A- and B-tubule connections in axonemes. Two recent studies suggested that CFAP77 orthologues are not present in the axoneme of *Trypanosoma brucei* [17,18], indicating that the mechanism underlying OJ maintenance differ between flagellated parasites and other commonly-used eukaryotic ciliary organisms.

The B-tubule nucleates at the surface of the A-tubule. An in vitro DMT assembly study suggested that removal of the C-terminal tail of A10 and A11 protofilaments allows the nucleation of a B-tubule from the B01 protofilament [19]. In *Cfap77*-KO mice, open B-tubules specifically at the OJ regions were observed in only approximately half of the DMTs, whereas many A- and B-tubules remained intact. Our study indicates that CFAP77 is not necessary for B-tubule formation but is necessary specifically for the stability of the OJ.

We found that the disconnection of A- and B-tubules was distributed mainly in DMTs 1, 4, 5, 6, and 9 in the sperm axonemes of *Cfap77*-KO mice. However, why certain DMTs are more prone to structural defects than others is still unknown. The asymmetric architecture of mammalian sperm axonemes has been revealed by in situ cryo-ET [20]. Each of the nine DMTs is decorated with a distinct combination of complexes, and uneven force distribution occurs during sliding. We suggest that DMTs 1, 4, 5, 6, and 9 in axonemes may suffer excessive force during beating and are more prone to break at weak OJ sites after the loss of CFAP77; however, further experiments are needed to clarify this.

High-resolution information on axonemal ultrastructures is critical for understanding their components and assembly [7–9]. Furthermore, structural biology could be combined with knockout animal models to provide in-cell structural insight into how deficiency of a protein in vivo disrupts the axoneme ultrastructure at the near-atomic level [8]. In a recent study, the sperm DMTs of *Tekt5*-KO mice were compared with those of their WT counterparts, providing an example of the application of in situ structural technologies directly to a knockout mouse strain [8]. However, *Tekt5*-KO mice are fertile, and their sperm motility is normal, prohibiting an analysis of the biological significance of tektin 5 in axoneme function. Our

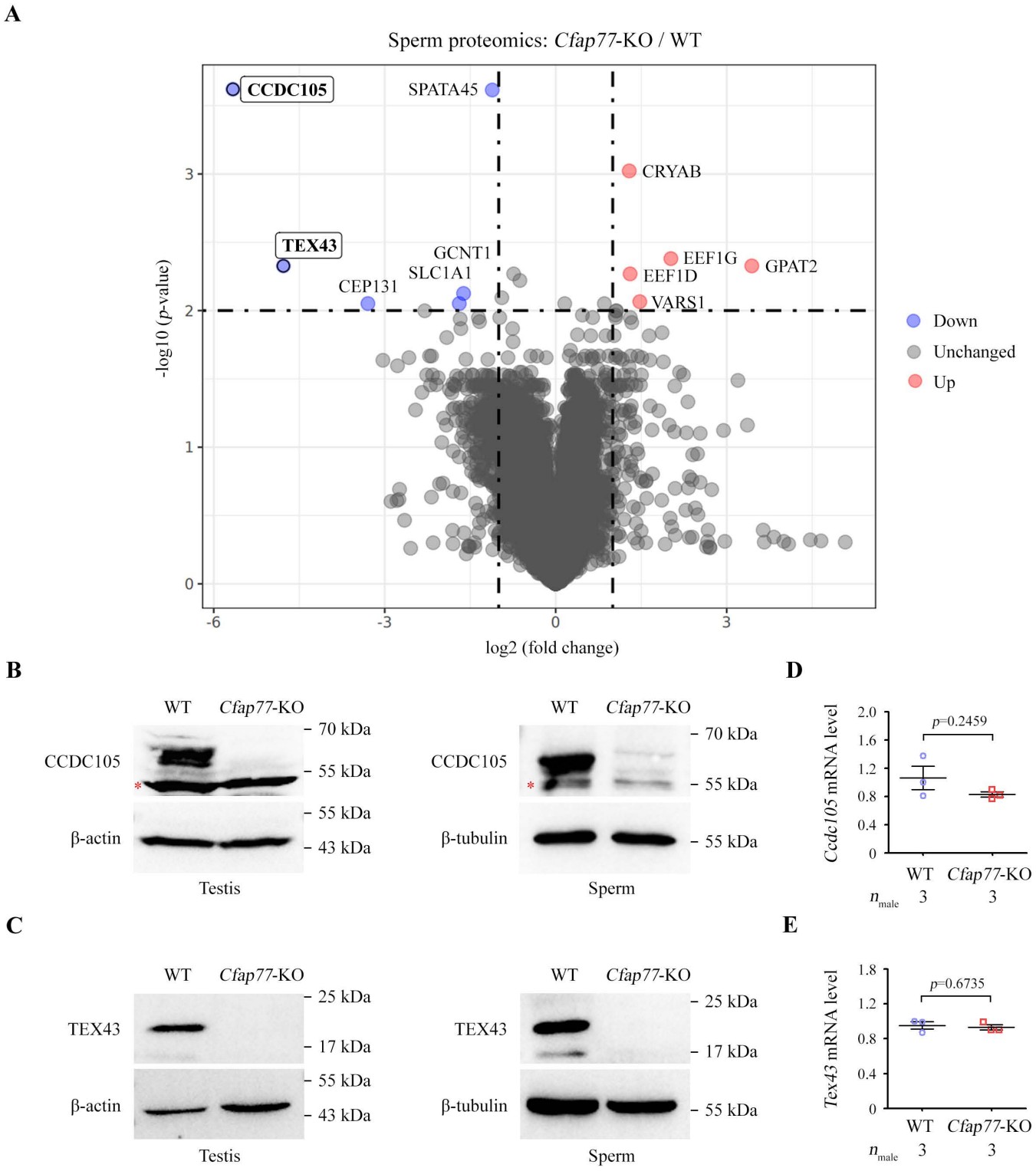

**Fig 5. Proteomic analysis of sperm from WT and *Cfap77*-KO mice. (A)** Quantitative proteomics of sperm protein lysates from WT mice and *Cfap77*-KO mice (*n* = 3 each group). Volcano plot showing the differentially expressed proteins. The X-axis and Y-axis represent the fold change (log2) and *p*-value (-log10), respectively. **(B)** Representative immunoblot of CCDC105 in the protein lysates of testes or sperm from WT mice and *Cfap77*-KO mice. β-actin or β-tubulin served as a loading control. Red stars indicate unspecific bands. **(C)** Expression of TEX43 in the protein lysates of testes or

sperm from WT mice and *Cfap77*-KO mice. β-actin or β-tubulin served as a loading control. **(D)** Relative mRNA level of *Ccdc105* in the testes of WT mice and *Cfap77*-KO mice, as revealed by qRT–PCR. Student *t* test; error bars represent the SEM ($n = 3$). **(E)** qRT–PCR results showing the relative mRNA level of *Tex43* in the testes of WT mice and *Cfap77*-KO mice. Student *t* test; error bars represent the SEM ($n = 3$). The data underlying the graphs shown in the figure can be found in S1 Data, in S2 Table and in the ProteomeXchange (dataset identifier PXD056128). Raw blot images can be found in S1 Raw Images.

current study further illustrates the advantages of the combination of structural biology and knockout animal experiments. Despite the critical role of CFAP77 in the connection of DMT A- and B-tubules, the molecular mechanism underlying the open DMT B-tubules at the OJ region is unknown. By applying in situ structural biology directly to *Cfap77*-KO mice, we observed the loss of the CFAP77-CCDC105-TEX43 subcomplex in the OJ region in situ in axonemal DMTs, which represents as an initial event precipitating the formation of open DMT B-tubules.

Research on mouse sperm DMTs [7–9] revealed that (i) CCDC105 forms a filament through head-to-tail assembly and that (ii) TEX43 and CFAP77 further interact with CCDC105 filaments to reinforce their linkage with the tubulin wall. Considering the complete loss of the CFAP77-CCDC105-TEX43 subcomplex in the sperm of *Cfap77*-KO mice, we suggest that the attachment of CCDC105 filaments to OJ regions is highly dependent on CFAP77. Without CFAP77, the CCDC105-TEX43 subcomplex is unable to localize to DMTs and is degraded in the cytoplasm of spermatids. The protein stability of CCDC105 and TEX43 may also be dependent on CFAP77, but further experiments are needed for verification.

CFAP77 is an evolutionarily conserved core OJ protein, whereas CCDC105 and TEX43 are two sperm-specific microtubule inner proteins. It has been recently reported that *Ccdc105*-KO mice exhibit no obvious effect on male reproduction [21] and that TEX43 is involved in the regulation of sperm motility but is also dispensable for male fertility [22]. Given that ultrastructure analyses (e.g., traditional TEM and cryo-EM/cryo-ET) are not performed in the sperm axonemes of these knockout mice, whether the loss of CCDC105 or TEX43 will partially disturb the DMT A- and B-tubule connections is unknown. The effects of CCDC105 and TEX43 deficiency on the expression and function of CFAP77 should also be studied. Regarding the phenotypes of these three knockout mice, the physiological role of CFAP77 in the axoneme and sperm motility seems to be much more important than its interaction with the proteins CCDC105 and TEX43. It would also be interesting to generate a *Cfap77* transgenic mice and breed them with *Cfap77*-KO mice to study whether the CFAP77-CCDC105-TEX43 subcomplex would be restored at the OJ regions and whether the phenotype of open DMT B-tubules, abnormal sperm motility and male infertility would be abrogated.

CFAP77 is predominantly expressed in the testes, but lower levels of CFAP77 can also be detected in other ciliary tissues, including the brain, lung, fallopian tube, and retina. With the exception of male sterility, *Cfap77*-KO mice do not display obvious symptoms of ciliopathy, including laterality abnormalities, hydrocephalus, or defects in respiratory cilia. However, disconnection of A- and B-tubules within DMTs also occurs in the ciliated cells of *Cfap77*-KO mice. Ultrastructural differences in DMT structures between sperm flagella and motile multiciliated epithelial cells [10] may account for their differential sensitivity to CFAP77 depletion. The epithelial cells in the ependyma, trachea, and nasal cavity are multiciliated, and some cilia with open DMT B-tubules may not generate obvious phenotypes. In contrast, sperm undergo long-distance migration with higher oscillation frequency; thus, the effect of axoneme instability may be more easily manifested in these cells. Indeed, a greater percentage of the axoneme exhibited open DMT B-tubules in sperm flagella (approximately 45%) than in other types of cilia (less than 30% in the trachea) in *Cfap77*-KO mice. After sperm hyperactivation, the phenotype of disconnected A- and B-tubules in the sperm axoneme is much more obvious (increasing to ~70%) in *Cfap77*-KO mice. We also cannot exclude the possibility that the effects of CFAP77 on other cilia may exist, but more precise investigations are needed.

Microtubule inner protein variant-associated asthenozoospermia (MIVA) is a subtype of asthenozoospermia and is characterized by impaired sperm motility without evident morphological abnormalities [7]. Given that the DMT A- and B-tubule connections are critical for the stability of sperm axonemes, deficiency in the CFAP77-CCDC105-TEX43

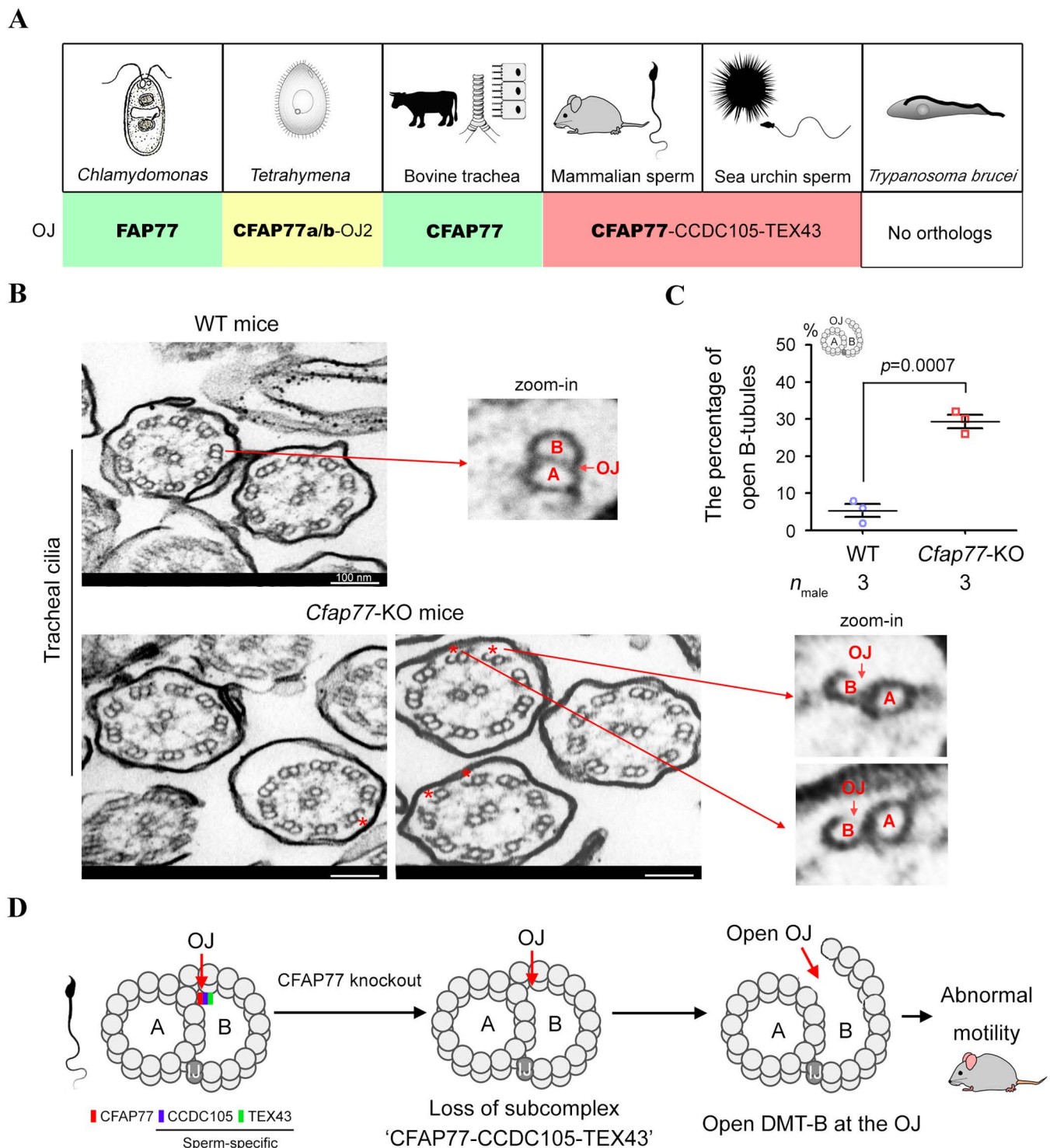

**Fig 6. CFAP77 is an evolutionarily conserved OJ component that is critical for the DMT A- and B-tubule connections in cilia. (A)** CFAP77 is present in different species, including *Chlamydomonas* (EMD-20631), *Tetrahymena* (EMD-29666), sea urchins (EMD-40619), bovines (EMD-24664), mice (EMD-35823) and humans (EMD-35810). There are no orthologues of CFAP77 in *Trypanosoma brucei*. In contrast, CCDC105 and TEX43 are sperm-specific CFAP77-binding proteins. **(B)** TEM analysis of axoneme structures in tracheal cilia from WT mice and *Cfap77*-KO mice. The red stars indicate the OJ region where open DMT B-tubules exist. A single DMT was zoomed in. Scale bars, 100 nm. **(C)** The percentage of axonemes in tracheal

cilia with open DMT B-tubules was calculated. For each mouse, 50 axonemes were randomly selected to examine the connection of A- and B-tubules. Student *t* test; error bars represent the SEM (*n* = 3). **(D)** Schematic diagram illustrating the connection of DMT A- and B-tubules by CFAP77: *Cfap77* knockout leads to the loss of the CFAP77-CCDC105-TEX43 subcomplex, the disconnection of A- and B-tubules specifically at the OJ regions, and ultimately, defective motility in mice. The data underlying the graphs shown in the figure can be found in S1 Data.

subcomplex may be a potential genetic cause of human male infertility with MIVA. Although dynein arms, radial spokes and nexin-dynein regulatory complexes can still attach to DMTs to form a complete axoneme structure with the CPC, the unstable connection of A- and B-tubules in mutants deficient in the CFAP77-CCDC105-TEX43 ternary complex leads to axoneme instability and the cumulative disruption of sperm progressive motility. Indeed, a homozygous mutation in *CCDC105* (c.G1042A/p.E348K) has been identified in infertile male patients with MIVA [7]. However, clinical evidence linking *CFAP77* and *TEX43* mutations to male infertility is still lacking. Whole-exome sequencing of a large cohort of infertile men, primarily with MIVA, will be useful for screening for any pathogenic mutations in *CFAP77* and *TEX43*.

In summary, our analyses using *Cfap77*-KO mice and in situ structural biology revealed that knockout of the core OJ protein CFAP77 leads to loss of the CFAP77-CCDC105-TEX43 subcomplex and the opening of DMT B-tubules at the OJ sites, defective sperm motility, as well as male infertility in mice. Our study not only provides insight into the molecular mechanism of A- and B-tubule connections within axonemal DMTs but also establishes a paradigm with which to study the axoneme by a combination of in situ structural biology and gene-edited animal models (and possibly human patient samples).

## Methods

### Generation of *Cfap77*-KO mice

The animal experiments were approved by the Animal Care and Use Committee of the College of Life Sciences, Beijing Normal University (CLS-AWEC-B-2023–001). The mouse *Cfap77* gene has 5 transcripts and is located on the chromosome 2. Exons 2 and 3 of the *Cfap77–202* (ENSMUST00000157048) transcript were selected as the knockout region. The generation of knockout mice was described in detail in our previous studies [23–25]. Briefly, mouse zygotes were coinjected with an RNA mixture of Cas9 mRNA (TriLink BioTechnologies, CA, USA) and sgRNAs. The injected zygotes were subsequently transferred into pseudopregnant recipients to obtain the F0 generation. DNA was extracted from the tail tissues of 7-day-old offspring, and PCR amplification was carried out with genotyping primers (S1 Table). A stable F1 generation was obtained by mating positive F0 generation mice with WT C57BL/6JG-pt mice. The gRNA sequence and Sanger sequence are illustrated in Fig 2A.

### Polyclonal antibody generation

The protocol followed for polyclonal antibody generation was described in detail in our previous study [24]. Briefly, recombinant full-length mouse CFAP77 (271 aa) and TEX43 (141 aa) were cloned and inserted into the pET-N-His-C-His vector (Beyotime, Shanghai, China) and then transfected into the ER2566 *E. coli* strain (Weidi Biotechnology, Shanghai, China). Protein expression was induced with 1 mM IPTG at 30°C overnight. After centrifugation, the bacterial pellet was resuspended in buffer (50 mM Tris-HCl pH 8.0, 200 mM NaCl), and the proteins were released by sonication. After further centrifugation, anti-His beads (Beyotime) were added to the supernatant, which was subsequently incubated overnight at 4°C. After washing, the recombinant protein was eluted with 250 mM imidazole (Beyotime). Recombinant protein was emulsified at a 1:1 ratio (v/v) with Freund's complete adjuvant (Beyotime) and administered subcutaneously to ICR female mice at multiple points. For the subsequent three immunizations, recombinant protein was emulsified with incomplete Freund's adjuvant (Beyotime) at an interval of 2 weeks. One week after the last immunization, blood was collected, and the serum was separated. The polyclonal antibodies used in this study were validated by overexpression and knockout/knockdown experiments (S10 Fig).

### Fertility testing

Adult *Cfap77*-KO male mice and their WT littermates ($n = 5$ each) were mated with WT C57BL/6J females (male:female = 1:2) for two months. The vaginal plugs of the mice were examined every morning. Female mice with vaginal plugs were fed separately and replaced by additional females to maintain the 1:2 male:female ratio. The number of pups per litter was recorded.

### In vitro fertilization (IVF)

Six-week-old ICR female mice were superovulated by the injection of 5 IU of pregnant mare serum gonadotropin (PMSG), followed by the injection of 5 IU of human chorionic gonadotropin (hCG) 48 h later. Sperm capacitation was performed for 50 min using TYH solution. Cumulus–oocyte complexes were obtained from the ampulla of the uterine tube at 14 h after hCG injection. Cumulus cells were removed by treatment with 350 μg/mL hyaluronidase for 10 min. To eliminate the zona pellucida, the oocytes were treated with 1 mg/mL collagenase for 8 min. The capacitated sperm were introduced into a drop containing cumulus-intact, cumulus-free, or zona pellucida-free oocytes at 37°C under 5% $CO_2$. Final concentrations of $2 \times 10^6$ sperm/mL and $2 \times 10^5$ sperm/mL were used for cumulus-intact/free IVF and zona pellucida-free IVF, respectively. After 6 h, the eggs were transferred to liquid drops of KSOM. Two-cell embryos were counted at 1 day postfertilization. All reagents were purchased from Aibei Biotechnology (Nanjing, China).

### Semen analysis

Sperm counts were determined using a fertility counting chamber (Makler, Israel) under a light microscope. Sperm motility was assessed using the application of a computer-assisted sperm analysis (CASA) system (SAS Medical, China). The sperm suspension was mounted on a glass slide, air-dried, and fixed with 4% paraformaldehyde (PFA) for 20 min at room temperature. The slides were stained with Papanicolaou solution (Solarbio, Beijing, China) and observed using a DM500 optical microscope (Leica, Germany).

### Transmission electron microscopy (TEM)

Samples were fixed with 2.5% (vol/vol) glutaraldehyde (GA) in 0.1 M phosphate buffer (PB) at 4°C. The samples were then washed four times in PB and first immersed in 1% (wt/vol) OsO4 and 1.5% (wt/vol) potassium ferricyanide aqueous solutions at 4°C for 2 h. After washing, the samples were dehydrated through a series of graded alcohol solutions into pure acetone. The samples were infiltrated in a graded mixture of acetone and SPI-PON812 resin, and then the pure resin was changed. Subsequently, the samples were embedded in pure resin with 1.5% BDMA, polymerized for 12 h at 45°C and 48 h at 60°C, cut into ultrathin sections (70 nm thick), and then stained with uranyl acetate and lead citrate for observation and imaging with a Tecnai G2 Spirit 120 kV (FEI) electron microscope. All reagents were purchased from Zhongjingkeyi Technology (Beijing, China).

### Western blotting

Proteins were extracted using RIPA lysis buffer containing 1 mM PMSF and 2% (v/w) protease inhibitor cocktail (Roche, Basel, Switzerland) on ice. The supernatants were collected following centrifugation at 12,000 × g for 10 min. The proteins were electrophoresed on 10% SDS–PAGE gels and transferred to PVDF membranes (GE Healthcare, USA). The blots were blocked in 5% milk and incubated with primary antibodies overnight at 4°C, followed by incubation with secondary antibody for 1 h. For primary antibodies, mouse anti-CFAP77 (formulated in-house, 1:1000), mouse anti-TEX43 (formulated in-house, 1:1000), and rabbit anti-CCDC105 (Proteintech, 24026–1-AP, 1:1000) were used. Mouse anti-β-actin (Abcam, ab8226, 1:1000), mouse anti-β-tubulin (Proteintech, a66240-1-Ig, 1:1000), or rabbit anti-vinculin (Proteintech, 26520–1-AP, 1:1000) served as internal controls. The secondary antibodies used were goat anti-rabbit IgG H&L (HRP)

(Abmart, M212115, 1:5000) and rabbit anti-mouse IgG H&L (HRP) (Abmart, M212131, 1:5000). The signals were evaluated using Super ECL Plus Western blotting Substrate and a Tanon-5200 Multi chemiluminescence imaging system (China).

## Quantification proteomics

Sperm were retrieved from the cauda epididymis of 10-week-old *Cfap77*-KO mice and WT mice ($n = 3$ each group) and $1 \times 10^7$ sperm from each group were used for quantification proteomics. Proteins were extracted from the sperm samples (a total of 6 samples) using 0.1 M Tris-HCl (pH 8.0), 0.1 M dithiothreitol (DTT), 4% SDS, 1 mM PMSF, and 2% (v/w) protease inhibitor cocktail (Roche, Basel, Switzerland), followed by sonication (20% amplitude, 10 pulses, three times) on ice. The supernatants were collected following centrifugation at $12,000 \times g$ for 20 min. The mass spectrometry protocol was described in detail in our previous studies [23–25]. Proteins with a cut-off point of a 2-fold change and a *p*-value less than 0.05 are listed in S2 Table. The mass spectrometry data have been deposited with the ProteomeXchange Consortium using the iProX partner repository with the dataset identifier PXD056128.

## Quantitative RT–PCR

Total RNA was extracted from the testes of WT mice and *Cfap77*-KO mice using an RNA Easy Fast Tissue/Cell Kit. The RNA was converted into cDNA with a FastKing One-Step RT–PCR Kit according to the manufacturer's instructions. The cDNAs were used as templates for subsequent real-time fluorescence quantitative PCR with RealUniversal Colour PreMix (SYBR Green). *Ccdc105* and *Tex43* mRNA expression was quantified according to the $2^{-\Delta\Delta Ct}$ method. Mouse *Actb* was used as an internal control. The primers used are listed in S3 Table. All kits were purchased from Tiangen Biotech (Beijing, China).

## Plasmid construction and cell culture

The full-length cDNAs encoding *CFAP77*, *CCDC105*, and *TEX43* were amplified by PCR; cloned and inserted into FLAG-, MYC-, or HA-tagged pCMV vectors; and confirmed by Sanger sequencing (Sangon Biotech, Shanghai, China). Mutant plasmids were constructed by using a QuickMutation Site-Directed Mutagenesis Kit (Beyotime, Shanghai, China). HEK293T cells (ATCC, NY, USA) were cultured at 37°C in a 5% $CO_2$ incubator with Dulbecco's modified Eagle's medium supplemented with 10% foetal bovine serum and 1% penicillin–streptomycin (Gibco, NY, USA).

## Coimmunoprecipitation (co-IP)

HEK293T cells were transfected with FLAG-, MYC-, and/or HA-tagged plasmids by Lipofectamine 3,000 (Thermo Fisher, CA, USA). Forty-eight hours after transfection, the cells were lysed with Pierce IP Lysis Buffer (Thermo Fisher) containing a 2% (v/w) protease inhibitor cocktail (Roche, Basel, Switzerland) for 30 min at 4°C and then centrifuged at $12,000 \times g$ for 10 min. The protein lysates were incubated overnight with FLAG antibody (Abmart, PA9,020, 2 μg) at 4°C. The lysates were then incubated with 20 μL of Pierce Protein A/G-conjugated agarose for 4 h at 4°C. The agarose beads were washed five times with Pierce IP Lysis Buffer and boiled for 5 min in $1 \times$ SDS loading buffer. Input and IP samples were analysed via Western blotting with an HRP-conjugated anti-FLAG antibody (Abmart, PA9,020, 1:1000), anti-MYC antibody (Abmart, M20019, 1:1000), or anti-HA antibody (Abmart, M20003, 1:1000).

## Sample preparation of mouse sperm axoneme

Freshly extracted sperm were centrifuged at $400 \times g$ (Thermo Scientific Legend Micro 17 R) for 5 min at 4°C. The precipitate per 100 μL of semen was carefully resuspended in 100 μL of precooled PBS and diluted 5.5-fold with PBS before use. The cryo-EM grid (Quantifoil R2/1, Au 200 mesh) was discharged for 60 s using Gatan Solarus. The sperm samples were quickly

frozen by vitrification with Leica EMGP. Samples diluted in 3 μL of PBS were immediately water absorbed at 100% relative humidity and 4°C for 4 s, and then the frozen samples were placed in a mixture of ethane and methane cooled to −195°C and stored in liquid nitrogen for cryo-FIB thinning. The cryo-FIB thinning strategy was described in our previous work [9].

### Cryo-ET tilt series collection

After the reduction of the cryo-FIB, the grid was mounted onto an autoloader in a Titan Krios G3 (Thermo fisher Scientific) 300 kV transmission electron microscope, equipped with a Gatan K2 direct electron detector (DED) and a BioQuantum energy filter. Tilt series were collected at a magnification of ×42,000, resulting in a physical pixel size of 3.4 Å in K2 DED and 3.4 Å in counting mode. Before data collection, the pretilt of the sample was determined visually, and the pretilt was set to 10° or −9° to match the predetermined geometry induced by loading grids. The total dose for each tilt was set to 3.5 electrons per angstrom squared and divided into 10 frames over a 1.2-s exposure, and the tilt angle was set to −9° pretilt of −66° to +51° or +10° pretilt of −50° to +67° in steps of 3°, resulting in 40 tilts and 140 electrons per tilt series. The slit width was set to 20 eV, zero-loss peaks were refined after the collection of each tilt series, and the nominal defocus was set from −1.8 to −2.5 μm. All tilt series used in this study were collected using a beam-to-image-shift facilitated acquisition scheme based on a dose symmetry strategy using a script developed in-house in SerialEM software [26–28].

### Data processing

After the data were collected, all the fractioned movies were imported into Warp for essential processing, including motion correction, Fourier binning by a factor of 1 for the counting mode frames, CTF estimation, masking of platinum islands or other high-contrast features, and tilt series generation [29]. Subsequently, AreTomo was used to automatically align the tilt series [29,30]. A visual inspection of the aligned tilt series in IMOD was conducted and any low-quality frames (such as those blocked by sample tables or grid bars, containing visible crystalline ice or showing significant jumps) were deleted to create the new tilt series in Warp. The new tilt series underwent a second round of AreTomo alignment. Then, according to the same criteria as in the first round, low-quality frames were again deleted. The new tilt series then underwent a third round of automatic alignment, continuing the process until no frames needed to be removed. After the alignment of the tilt series, those tilt series that were shorter than 30 frames or failed were not further processed [30,31]. All remaining alignment parameters for the tilt series were passed back to Warp, and an initial layer map reconstruction was performed in Warp with a pixel size of 27.2 Å. Among the 174 tomograms, the filament picking tool in Dynamo was used to manually pick DMT particles from 100 of all the tomograms. One hundred sets of DMT particles were obtained by selecting the starting and ending points of each DMT fibre and separating each cutting point by 8-nm along the fibre axis, 100 sets of DMT particles were obtained [32]. The 3D coordinates and two of the three Euler angles (except for in-plane rotation) were automatically generated by Dynamo and then transferred back to Warp to export subtomograms [29,32,33]. The subtomograms were refined using the ABTT package to transform the RELION star file and Dynamo table file and Dynamo and/or RELION to jointly generate a mask [34,35]. First, all the particles were reconstructed into a box size of $48^3$ voxels with a pixel size of 27.2 Å, and all the extracted particles were directly averaged and low-pass filtered at 80 Å to generate an initial reference. Then, 3D classification with K = 1 was performed under the constraints of the first two Euler angles (—sigma_tilt 3 and —sigma_psi 3 in RELION), and 3D automatic refinement was performed after 25 iterations. After alignment, the particles were manually cleaned in ChimeraX-1.6 [36], and then these aligned parameters were transferred back to Warp to export the subtomograms with a box size of $84^3$ voxels and a pixel size of 13.6 Å. The particles were automatically refined in RELION and these aligned parameters were transferred back to Warp to export the subtomograms with a box size of $128^3$ voxels and a pixel size of 6.8 Å. Then, the subtomograms were automatically refined in RELION. After any duplicate particles in Dynamo were removed, they were automatically refined in M with a final resolution of 24 Å [37]. On the basis of the previous atomic model of DMT in WT mice (PDB: 8I7O), we deleted the models of CFAP77, CCDC105, and TEX43 in ChimeraX-1.6 to obtain the DMT model of *Cfap77*-KO mouse sperm [9].

## Analysis and statistical analyses

All tests were performed and analysed by an experimenter who was blinded to the genotype. The data are presented as the means ± standard errors of the means (SEMs) and were analysed via GraphPad Prism version 5.01 (GraphPad Software). Student $t$ test (unpaired, two-tailed) was used for the statistical analyses.

## Supporting information

**S1 Fig. Interactions among CFAP77, CCDC105, and TEX43.** (A) HA-tagged CFAP77 could be immunoprecipitated by FLAG-tagged CCDC105 in HEK293T cells. Vinculin served as the internal control. (B) HA-tagged TEX43 could also be immunoprecipitated by FLAG-tagged CCDC105 in HEK293T cells. β-actin served as the internal control. (C) No interaction between MYC-tagged CFAP77 and FLAG-tagged TEX43 in HEK293T cells. β-actin served as the internal control. These coimmunoprecipitation results were consistent with the structural data showing that CCDC105 interacts with both CFAP77 and TEX43 but that there is no interaction between CFAP77 and TEX43. Raw blot images can be found in S1 Raw Images.
(TIF)

**S2 Fig. Analysis of the reproductive phenotypes of *Cfap77*-KO mice.** (A) Representative histological morphology of testis sections from *Cfap77*-KO mice and WT mice determined by periodic acid-schiff (PAS) staining. Spc, spermatocytes; Rs, round spermatids; Es, elongating spermatids. Scale bars, 50 μm. (B) Staining of PNA (peanut agglutinin)-TRITC to reveal acrosomal formation in testis sections from *Cfap77*-KO mice and WT mice. Nuclei were stained with DAPI. (C) Representative histological morphology of the cauda epididymis of *Cfap77*-KO mice and WT mice, as determined by haematoxylin and eosin (H&E) staining.
(TIF)

**S3 Fig. Detailed analysis of open DMT B-tubules of sperm flagella in *Cfap77*-KO mice.** (A) Transmission electron microscopy analysis of acrosome (ACR), mid-piece (Mp) and principal piece (Pp) of the tails of WT mice and *Cfap77*-KO mice. Nu, nucleus; MS, mitochondrial sheath; FS, fibrous sheath; CPC, central pair complex; DMT, doublet microtubule; ODF, outer dense fibre. Scale bars, 100 nm or 500 nm. (B) Number of each DMT (1–9) exhibiting the open B-tubules. A total of 30 axonemes with open DMT B-tubules in *Cfap77*-KO sperm were counted. (C) The percentage of axonemes with open DMT B-tubules was calculated between the Mp and Pp of sperm flagella in *Cfap77*-KO mice. Student $t$ test; error bars represent the SEM ($n = 3$). The data underlying the graphs shown in the figure can be found in S1 Data.
(TIF)

**S4 Fig. Cryo-FIB milling of sperm from *Cfap77*-KO mice.** (A) Inspection of frozen sperm on the grid. (B) Inspection of frozen sperm on the grid after FIB milling. The thin lamellae were used for data collection. (C) In-cell structural determination of sperm axonemal DMTs from *Cfap77*-KO mice. Side and transverse sectional views of mouse sperm axonemes are shown in the tomogram slices.
(TIF)

**S5 Fig. Data processing of sperm DMTs from *Cfap77*-KO mice.** The pixel sizes at different binning levels are indicated in angstroms per pixel (Å/px for short). The half-map Fourier shell correlation (FSC) plot is shown.
(TIF)

**S6 Fig. Sequence alignment analysis of CFAP77.** Using the ClustalW and ESPript websites, we revealed that CFAP77 was remarkably conserved among humans, bovines, mice, sea urchins, *Tetrahymena*, and *Chlamydomonas*. The amino acid sequences of CFAP77 were obtained from UniProt: human (Q6ZQR2), bovine (A0A3Q1LJD6), mouse (A0A087WRI3), sea urchin (A0A7M7TG06), *Tetrahymena* (Q22WR6), and *Chlamydomonas* (A8IB22).
(TIF)

**S7 Fig. Cryo-EM/cryo-ET structural models of CFAP77, CCDC105, and TEX43.** Structural models of the CFAP77-CCDC105-TEX43 subcomplex or CFAP77 at the OJ regions of axonemes from various species and tissues are presented, with a vertical section showing a 48 nm length of the DMTs. From top to bottom, the models include mouse sperm (PDB: 8IYJ), bovine sperm (PDB: 8OTZ), sea urchin sperm (PDB: 8SNB), human respiratory epithelium (PDB: 7UNG), pig fallopian tube (PDB: 9CPB), pig brain ventricles (PDB: 9CPC), and *Tetrahymena* (PDB: 8G3D). (TIF)

**S8 Fig. Expression information for the *CFAP77*/*Cfap77*, *CCDC105*, and *TEX43* mRNAs.** (A) *CCDC105* mRNA was restricted to the testes in humans. These data were obtained from the Human Protein Atlas (HPA) database (https://www.proteinatlas.org/ENSG00000160994-CCDC105/tissue). (B) *TEX43* mRNA was also restricted to the testes in humans. These data were obtained from the HPA database (https://www.proteinatlas.org/ENSG00000196900-TEX43/tissue). (C) Human *CFAP77* mRNA was predominantly expressed in the testes but was also present in other ciliary tissues, including the fallopian tube, retina, brain, and lung. These data were obtained from the HPA database (https://www.proteinatlas.org/ENSG00000188523-CFAP77/tissue). (D) According to the results of quantitative real-time PCR, *Cfap77* mRNA was highly expressed in mouse testes and epididymis, but was also expressed in other tissues. Student $t$ test; error bars represent the SEM ($n = 3$). The data underlying the graphs shown in the figure can be found in S1 Data. (TIF)

**S9 Fig. Scanning electron microscopy analysis of cilia in *Cfap77*-KO mice.** (A) Representative scanning electron microscopy images of brain ependymal cilia from *Cfap70*-KO mice and WT mice. Scale bar, 5 µm. (B) Representative scanning electron microscopy images of tracheal cilia from *Cfap70*-KO mice and WT mice. Scale bar, 10 µm. (C) Representative scanning electron microscopy images of nasal cilia from *Cfap70*-KO mice and WT mice. Scale bar, 5 µm. (TIF)

**S10 Fig. Antibody verification.** (A) Overexpression verification. HEK293T cells were transfected with CFAP77-, CCDC105-, or TEX43-tagged plasmids; both tag-antibodies and their antibodies recognized a specific band at the same molecular size. (B) Knockout validation. CFAP77 was absent in sperm samples of *Cfap77*-KO mice. (C) siRNAs targeting *Ccdc105* and *Tex43* reduced their signals in HEK293T cells transfected with CCDC105- or TEX43-tagged plasmids, respectively. The siRNA sequences were listed as follows. NC siRNA: UUCUCCGAACGUGUCACGUUACGUGACACGUUCGGAGAATT; siRNA targeting *Ccdc105*: CGUGUGCUAAGGCCUUGUUUAACAAGGCCUUAGCACACGTT; siRNA targeting *Tex43*: GGUGGGACGAUAUUCACUUUAAGUGAAUAUCGUCCCACCTT. Raw blot images can be found in S1 Raw Images. (TIF)

**S1 Video. Sperm motility of WT and *Cfap77*-KO mice.** (MP4)

**S1 Table. Primers for mouse genotyping.** (DOCX)

**S2 Table. Proteins with a cut-off point of a 2-fold change and a *p*-value less than 0.05 were included.** (XLSX)

**S3 Table. Primers for qRT–PCR.** (DOCX)

**S4 Table. Cryo-ET data collection and data processing.** (DOCX)

**S1 Data.  Numerical values underlying all graphs in the main body and supporting information.**
(XLSX)

**S1 Raw Images.  Uncropped version of all Western Blot images in the main body and supporting information.**
(PDF)

## Acknowledgments

We would like to thank the Center for Biological Imaging (CBI), Institute of Biophysics (IBP), Chinese Academy of Science (CAS) for their support with TEM, specimen vitrification, cryo-FIB milling, and cryo-ET data collection. We also acknowledge the Experimental Technology Center for Life Sciences, Beijing Normal University, for providing the experimental platform.

## Author contributions

**Conceptualization:** Su-Ren Chen.

**Data curation:** Lan Xia, Guo-Liang Yin, Yu Long.

**Funding acquisition:** Fei Sun, Bin-Bin Wang, Yun Zhu, Su-Ren Chen.

**Investigation:** Su-Ren Chen.

**Methodology:** Lan Xia, Guo-Liang Yin, Yu Long.

**Supervision:** Bin-Bin Wang, Yun Zhu, Su-Ren Chen.

**Writing – original draft:** Su-Ren Chen.

**Writing – review & editing:** Fei Sun, Bin-Bin Wang, Yun Zhu.

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
