## [Editor Report · Decision Letter 0]

26 Apr 2025

Dear Dr Chen,

Thank you for submitting your manuscript entitled "Insight into the connection of A and B tubules within axonemal microtubule doublets of cilia and flagella" for consideration as a Research Article by PLOS Biology.

Your manuscript has now been evaluated by the PLOS Biology editorial staff as well as by an academic editor with relevant expertise and I am writing to let you know that we would like to send your submission out for external peer review.

Once your full submission is complete, your paper will undergo a series of checks in preparation for peer review. After your manuscript has passed the checks it will be sent out for review. To provide the metadata for your submission, please Login to Editorial Manager (https://www.editorialmanager.com/pbiology) within two working days, i.e. by Apr 29 2025 11:59PM.

Kind regards,

Ines

--

Ines Alvarez-Garcia, PhD

Senior Editor

PLOS Biology

---

## [Decision Letter · Decision Letter 1]

2 Jun 2025

Dear Dr Chen,

Thank you for your patience while your manuscript entitled "Insight into the connection of A and B tubules within axonemal microtubule doublets of cilia and flagella" was peer-reviewed at PLOS Biology. The manuscript has now been evaluated by the PLOS Biology editors, an Academic Editor with relevant expertise, and by three independent reviewers.

The reviews are attached below. As you will see, the reviewers find the conclusions interesting and novel, but they also raise several issues that would need to be addressed before we can consider the manuscript for publication. Reviewer 1 thinks that the quantification of fertility estimation needs to be improved to be significant, and suggests a different strategy to identify particles with a conventional doublet microtubule architecture and recover additional states to confirm the TEM data. This reviewer also asks for a table of cryo-ET data collection and processing statistics that follow the convention in the field, and raises several points that should be clarified. Reviewer 2 mentions that it should be confirmed if fertility reduction is associated with motility aberrations or sperm abnormalities, cumulus-free and Zona pellucida-free IVF. Reviewer 3 has concerns regarding the statistical analysis and suggests increasing the number of animals analysed or provide a clear rationale for the chosen sample size. This reviewer also requests several clarifications on the methods used and clarification on the way some of the data are reported, and suggests co-IP experiments with TEX43 to analyse protein interactions and strengthen the conclusions.

In light of the reviews, we would like to invite you to revise the work to thoroughly address the reviewers' reports. Given the extent of revision needed, we cannot make a decision about publication until we have seen the revised manuscript and your response to the reviewers' comments. Your revised manuscript is likely to be sent for further evaluation by all or a subset of the reviewers.

**IMPORTANT - SUBMITTING YOUR REVISION**

3. Resubmission Checklist

a) *PLOS Data Policy*

b) *Published Peer Review*

d) *Blurb*

Please also provide a blurb which (if accepted) will be included in our weekly and monthly Electronic Table of Contents, sent out to readers of PLOS Biology, and may be used to promote your article in social media. The blurb should be about 30-40 words long and is subject to editorial changes. It should, without exaggeration, entice people to read your manuscript. It should not be redundant with the title and should not contain acronyms or abbreviations. For examples, view our author guidelines: https://journals.plos.org/plosbiology/s/revising-your-manuscript#loc-blurb

Sincerely,

Ines

--

Ines Alvarez-Garcia, PhD

Senior Editor

PLOS Biology

Reviewers' comments

Rev. 1:

Xia, Yin and colleagues address the importance of CFAP77 for doublet microtubule stability in mouse sperm. The demonstrate that genetic loss of CFAP77 causes sperm motility defects, male infertility, and B tubules open at the outer junction. They apply in-situ cryo-ET to show that loss of CFAP77 also causes loss of CCDC105 and TEX43. Overall, this is a convincing study with valuable findings that deserves publication once the following issues are resolved:

1. The fertility measurements are statistically underpowered (n=3). Recommendations are that at least 4 litters are needed to provide reliable and robust estimates of fertility.

2. The multiple mentions that it is a novel approach or that this the first study to combine in situ structural biology and knockout mice is incorrect and unnecessary.

3. The section on CFAP47 in the introduction is an unnecessary distraction (in addition to countering their assertion that this paper is the first to combine cryo-ET and knockout mice)

4. The failure of the cryo-ET approach to resolve classes in which the OJ is "open" suggests that the processing strategy may have been suboptimal and biased towards identifying those particles with a conventional doublet microtubule architecture. If a different strategy is used (e.g. initial processing ignoring the B tubule, followed by classification on the B tubule alone), can the authors recover additional states more consistent with the TEM data?

5. Whether SPACA9/EUKUR/TEKT5 are absent or not from the Cfap77-KO doublet microtubules is not clearly addressed in the Results. A relatively low resolution 8-nm map may be insufficient to identify all the structural changes.

6. Why does the mass spectrometry volcano plot (Fig. 5) have no data points between +1 and -1 log2(fold change)? All statistically significant changes should be labeled (especially as there are so few).

7. It is unclear how the difference map was created. Ideally, it should be generated by subtracting reconstructions of WT and Cfap77-KO from similar numbers of particles at similar resolutions and with the same periodicities.

8. Few details are provided as to how the TEM data were quantified. Was this analysis blind to genotype?

9. The manuscript should include a table of cryo-ET data collection and processing statistics, following the convention in the field.

10. In the TEM images, is it possible to quantify if the open B tubules still have 10 protofilaments (as drawn in the proposed model)?

11. The phylogenetic distribution of CFAP77 should be addressed more clearly. For example, two recent structural studies have shown that CFAP77 homologs are not present in trypanosomatids.

Minor comments.

Past and present tenses are used inconsistently in the Methods and figure legends, e.g. in the Data Processing section and Figure 1.

Please provide more sample preparation details for the "Quantitative proteomics" section. For example, how many sperm cells were used?

The acronym SEM is used to mean both scanning electron microscopy and standard error.

The Discussion should clearly state that the evidence shows CFAP77 is not necessary for B tubule formation but only for its stability.

L48. It is the dynein tails that associate with the A tubule, and the heads that associate with the neighboring B tubule. The text currently has the order wrong. It is usual to refer to the N-DRC as having a "tail".

L54. To be inclusive and comprehensive, studies of bovine sperm structure should be added.

L57. It should be made clear that CFAP77 is not the only protein present at the OJ, connecting A and B tubules.

L65. "Understating" should be "understanding"

L74. "knockout mice were male infertile" should be "male knockout mice were infertile"

L98. "glutarnine" should be "glutamine"

L122. Unnecessary to mention this is the "first" Cfap77-KO mouse strain.

L137. "focusing on axoneme" should be "focusing on the axoneme"

L155. All instances of "perfectly consistent" should be replaced with "consistent".

L156. What is meant by "expression evidence"? Structural data from WT doublet microtubules seems more appropriate to reference.

L159. "frozen" should be "froze"

L168. "This sperm DMTs" should be "The sperm DMTs"

Fig. 1. It would be helpful to show a zoomed in cross-sectional view of the OJ, sliced appropriately to see the arrangement of CFAP77/CCDC105/TEX43.

Fig. 1. In the zoomed-in panels, it would be helpful to show the interacting residues, e.g. P110-H114, in a slightly different shade of color. Stick representation could be used to show the residues mutated in panels E and F.

Fig. 5 "unchange" should be "unchanged"

Fig. 6B. Please also provide zoomed-in images of doublet microtubules from WT mice tracheal cilia.

Fig. 6C - the y axis shows percent rather than ratio.

Fig. S4. Please give extract particle numbers.

Fig. S6 A. The color-coding of the multiple-sequence alignment is subjective. The alignment might also benefit from additional species.

Fig. S6 B. Why show AlphaFold models, when the experimentally determined structures could be compared?

Please combine the two supplementary movies so that they are side-by-side. This facilitates comparison of the phenotypes.

Rev. 2:

The authors Xia et al., are studying poorly characterised gene/protein Cfap77 and demonstrate its critical role in murine fertility by uncovering the mechanism of its action by using a knock-out mouse model and state-of-the-art Cryo-electron tomography.

-The research question is clearly formulated, the experiments are specific and well-designed: e.g., mutating specific a.a. in Cfap77 instead of deleting the entire gene to validate the interactions in the ternary subcomplex (CFAP77-CCDC105-TEX43).

-The localization of Cfap77 to the microtubular doublet at the outer junction of A and B tubules was already reported earlier (in Tetrahymena https://doi.org/10.1038/s41467-023-37868-0, see urchin, bovine https://doi.org/10.1016/j.cell.2023.05.026); however, its phenotype and function mechanism in mammals is not known.

-In a nutshell, experiments have logical flow and validation with included controls, they demonstrate Cfap77 central role in male fertility in association with the loss of three proteins (CFAP77-CCDC105-TEX43) from the DMT outer junction that way providing the functional mechanism. All in all, technically study is sound and provides novel insights into the previously unreported function of the Cfap77 gene. However, the study would benefit from data representation and discussion polishing and language revision.

Some specific points to consider:

-IVF was done only with the cumulus-intact oocytes. To confirm if fertility reduction is associated with motility aberrations or sperm abnormalities, cumulus-free and Zona pellucida-free IVF should be done. It is likely that ZP-free oocytes could be fertilized in as wild types which would be beneficial information when planning ART for human cases.

-Line 129: 'Collectively, the animal studies indicate that CFAP77 is critical for progressive sperm motility and male fertility in mammals.'

This is likely too bold a statement since data is only from the mouse, given this, it only suggests importance in mammals but does not prove.

-Researchers analyzed duplet microtubules with open B microtubules in Cfap77-KO in ependymal, tracheal, and nasal cilia. It would be useful to show the expression levels of Cfap77 by RTqPCR or Western blot, at least in these tissues, or even a broader selection of ciliated tissues in comparison to testes and epididymis, where phenotype is prominent.

-Line 236: 'A combination of knockout animal studies with structural biology technologies is one of the cutting-edge trends in cilia biology: functional studies are essential to confirm the physiological roles of axonemal proteins and structural biology can further provide in-cell structural insight of how knocking out a protein disrupts the axoneme ultrastructure at near-atomic level.' Requires citations.

-Line 243: 'Our current study represents a good example to combine structural biology and knockout animals.' Liguistuly not sound.

-Line 247: 'Moreover, we also found that SPACA9 and ENKUR were missing in the B-tubule, and tilted Tektin5 (Tektin5-5) protein was absent in A-tubule lumen of sperm DMTs from Cfap77-KO mice.' This sentence refers to results that are not presented in the results section and are out of context as such.

-Line: 'From the perspective of the current study, we consider that CFAP77 is the evolutionarily conserved core OJ protein residing between A11-A12 and B01-B02, while CCDC105 and TEX43, two sperm-specific MIPs, may function to join CFAP77 molecules into a more secure footing for DMTs A-B tubule connection in sperm flagella.' In other parts of the text, authors discuss CFAP77 as an evolutionarily conserved protein, and here they write that 'we consider that CFAP77 is the evolutionarily conserved', why just 'consider'?

-How was the specificity of in-house derived antibodies validated (anti-CFAP77, mouse anti-TEX43 )?

-Line 402: cDNA encoding CFAP77, CCDC105, and TEX43 Italic

- Special note, I am not an expert on Cryo-electron tomography methodology, so I cannot provide specific comments on this.

Rev. 3:

In their study Xia et al. investigate the role of the outer junction CFAP77-CCDC105-TEX43 complex, previously identified in their cryo-EM analysis (Tai et al., 2023). Through alanine mutagenesis and co-IP experiments, the authors pinpointed key residues essential for complex formation. They generated a Cfap77 KO mouse model and demonstrated that the mice are infertile due to impaired sperm motility, likely resulting from disrupted outer junction architecture, as shown by electron microscopy. Cryo-EM/ET and mass spectrometry analyses confirmed that loss of CFAP77 leads to the absence of CCDC105 and TEX43. Interestingly, although CFAP77 is also expressed in other tissues where it similarly induces an open outer junction, no additional phenotypes were observed.

Overall, the manuscript is well written, and the narrative flows logically. The study presents compelling in vivo evidence that the stability of the outer junction structure is mediated by a defined protein complex, with CFAP77 as a central component. These findings provide significant insight into the molecular determinants of axonemal stability by linking specific structural components to ultrastructural integrity and motility.

My main concern lies with the statistical reporting. In several key fertility-related assays, including litter size, IVF, and sperm count/motility, the use of n = 3 biological replicates appears insufficient to support the conclusions with statistical confidence. I strongly recommend increasing the number of animals analyzed or, at the very least, providing a clear rationale for the chosen sample sizes. Additionally, the number of cells/sperm assessed should be explicitly stated (Fig. 2E, H, I). An n of 3 in these assays limits the power of the comparisons and may compromise the robustness of the conclusions.

I also have concerns regarding the use of homemade primary antibodies. Specifically, the predicted molecular weight of CCDC105 is approximately 50 kDa, which raises the possibility that the lower band in Figure 5B corresponds to CCDC105, rather than the upper band as currently indicated. While this point requires clarification, the mass spectrometry data does provide complementary support for the authors findings.

Minor Comments:

* It would be beneficial to include co-IP experiments with TEX43 to investigate protein interactions and to strengthen the authors claim.

* It would be interesting to discuss why certain DMTs are more prone to structural defects than others.

* Abstract + title: The authors use "DMT" as the abbreviation for "doublet microtubules" throughout the manuscript. To ensure consistency, they should revise the title and abstract to use the term "doublet microtubules" instead of "microtubule doublets."

* Usually hyphen is used between A/B tubule → "A-tubule" and "B-tubule".

* In their Intro the authors mention that CFAP20 is found in C. elegans (and zebrafish). It would be helpful to clarify that CFAP20 is also present in non-motile cilia, as C. elegans lack motile cilia.

* Page 5, Line 97: The phrase "relatively independent subcomplex" should be clarified. Specifically, do the authors mean that this complex does not physically or functionally interact with other microtubule inner proteins (MIPs)?

* Figure 5: It would be helpful if the authors could include information about proteins that were upregulated following Cfap77 knockout.

---

## [Decision Letter · Decision Letter 2]

3 Sep 2025

Dear Dr Chen,

Thank you for your patience while we considered your revised manuscript entitled "Insight into the connection of A- and B-tubules within axonemal doublet microtubules of cilia and flagella" for publication as a Research Article at PLOS Biology. This revised version of your manuscript has been evaluated by the PLOS Biology editors, the Academic Editor and the three original reviewers.

Based on the reviews, we are likely to accept this manuscript for publication, provided you satisfactorily address the remaining minor points raised by Reviewers 1 and 2. Please also make sure to address the following data and other policy-related requests stated below my signature.

In addition, we would like you to consider a suggestion to improve the title:

"The core outer junction protein CFAP77 connects A- and B-tubules within doublet microtubules of cilia and flagella"

We expect to receive your revised manuscript within two weeks.

*Published Peer Review History*

*Press*

Sincerely,

Ines

--

Ines Alvarez-Garcia, PhD

Senior Editor

PLOS Biology

Fig. 2C-F, H-I; Fig. 3C; Fig. 5A, D, E; Fig. 6C; Fig. S3B, C; Fig. S5 and Fig. S8A-D

Please also ensure that figure legends in your manuscript include information ON WHERE THE UNDERLYING DATA CAN BE FOUND, and ensure your supplemental data file/s has a legend.

In addition, please make sure that the data deposited at EMDB under accession code EMD-63176.

CODE POLICY

Thank you for providing the raw gels. Please note that the label of Fig. S1C gel is wrong and says again S1B, so it should be corrected.

Reviewers' comments

Rev. 1:

The authors have addressed all my concerns. The revised manuscript is clearly written and ready for publication.

During reading I noticed only two sentences were the authors might want to consider changes:

L26. Please remove "core" - it is unnecessary and potentially misleading as CFAP77 is not found in all doublet microtubules.

L55. "explored" should be "determined"

Rev. 2:

I have reviewed the response to my criticism and can conclude that it was appropriately addressed. However, the response letter contained incorrect information about the actual image numbers in the manuscript versus revised text: e.g. S7D->S8D; new Fig. S8B->S10B. After revising the manuscript text, it seems to be sound.

Rev. 3:

The authors have addressed all my concerns. The manuscript has improved significantly and I recommend it for publication in PLOS Biology.

---

## [Editor Report · Decision Letter 3]

30 Sep 2025

Dear Dr Chen,

Thank you for the submission of your revised Research Article entitled "The core outer junction protein CFAP77 connects A- and B-tubules within doublet microtubules of cilia and flagella" for publication in PLOS Biology. On behalf of my colleagues and the Academic Editor, Dagmar Wachten, I am delighted to let you know that we can in principle accept your manuscript for publication, provided you address any remaining formatting and reporting issues. These will be detailed in an email you should receive within 2-3 business days from our colleagues in the journal operations team; no action is required from you until then. Please note that we will not be able to formally accept your manuscript and schedule it for publication until you have completed any requested changes.

PRESS

Sincerely, 

Ines

--

Ines Alvarez-Garcia, PhD

Senior Editor

PLOS Biology
